# FDG-PET/CT for Response Monitoring in Metastatic Breast Cancer: Today, Tomorrow, and Beyond

**DOI:** 10.3390/cancers11081190

**Published:** 2019-08-15

**Authors:** Malene Grubbe Hildebrandt, Jeppe Faurholdt Lauridsen, Marianne Vogsen, Jorun Holm, Mie Holm Vilstrup, Poul-Erik Braad, Oke Gerke, Mads Thomassen, Marianne Ewertz, Poul Flemming Høilund-Carlsen

**Affiliations:** 1Department of Nuclear Medicine, Odense University Hospital, 5000 Odense, Denmark; 2Department of Clinical Research, University of Southern Denmark, 5230 Odense, Denmark; 3Centre for Innovative Medical Technology, Odense University Hospital, 5000 Odense, Denmark; 4Department of Nuclear Medicine, Lillebaelt Hospital, 7100 Vejle, Denmark; 5Department of Oncology, Odense University Hospital, 5000 Odense, Denmark; 6Department of Clinical Genetics, Odense University Hospital, 5000 Odense, Denmark

**Keywords:** precision oncology, FDG-PET/CT, PERCIST, metastatic breast cancer

## Abstract

While current international guidelines include imaging of the target lesion for response monitoring in metastatic breast cancer, they do not provide specific recommendations for choice of imaging modality or response criteria. This is important as clinical decisions may vary depending on which imaging modality is used for monitoring metastatic breast cancer. FDG-PET/CT has shown high accuracy in diagnosing metastatic breast cancer, and the Positron Emission Tomography Response Criteria in Solid Tumors (PERCIST) have shown higher predictive values than the CT-based Response Evaluation Criteria in Solid Tumors (RECIST) for prediction of progression-free survival. No studies have yet addressed the clinical impact of using different imaging modalities or response evaluation criteria for longitudinal response monitoring in metastatic breast cancer. We present a case study of a patient with metastatic breast cancer who was monitored first with conventional CT and then with FDG-PET/CT. We retrospectively applied PERCIST to evaluate the longitudinal response to treatment. We used the one-lesion PERCIST model measuring SULpeak in the hottest metastatic lesion on consecutive scans. This model provides a continuous variable that allows graphical illustration of disease fluctuation along with response categories. The one-lesion PERCIST approach seems able to reflect molecular changes and has the potential to support clinical decision-making. Prospective clinical studies addressing the clinical impact of PERCIST in metastatic breast cancer are needed to establish evidence-based recommendations for response monitoring in this disease.

## 1. Introduction

Breast cancer is the most frequent malignant disease among women. In developed countries, more than 90% are diagnosed with operable breast cancer, while 5–10% have advanced or metastatic disease at diagnosis. The prognosis is generally good, with 10-year survival rates of about 80% [1]. Survival with metastatic cancer is improving along with the rapid development of new treatments, such as HER2-targeting therapies [2]. Breast cancer is thus emerging as a chronic disease, where survivors can live for long periods of time.

In randomized clinical trials of new drugs, response to therapy is monitored using the New Response Evaluation Criteria in Solid Tumors (RECIST) approach [3], but current guidelines do not provide specific recommendations for response monitoring for patients with metastatic breast cancer outside clinical trials [4]. There is limited evidence supporting the clinical use of molecular imaging with 2-deoxy-2[18F] fluoroglucose-positron emission tomography/computed tomography (FDG-PET/CT) and the related PET Response Criteria in Solid Tumors (PERCIST) [5,6] for response monitoring in metastatic breast cancer. Due to the heterogeneity of breast cancer within and between patients, response monitoring has become a complex area, and specialized knowledge of the metastatic process and the molecular basis for response prediction and monitoring is required to enable the guidelines to be updated.

In this perspective paper, we address the potential of using PERCIST for longitudinal response monitoring in metastatic breast cancer and illustrate this with a case study. We reflect on the advantages of using PERCIST and suggest ways to improve and innovate the approach based on our experiences of using FDG-PET/CT for response monitoring in metastatic breast cancer. In addition, we relate advancements in molecular imaging technology to current knowledge about the biology of metastases and the evolving field of precision oncology.

## 2. Metastasis, Progression, and Precision Oncology

Treatment recommendations for primary breast cancer are based on the pathological molecular profile of the cancer’s dominant cell clone [7]. Primary tumors are often heterogeneous, however, and may present with minor components of differing tumor cell types, i.e., cells with differing molecular profiles [8]. When a cancer spreads to other tissues, the metastases can be of a different type to the primary cancer, and sometimes the metastases are even different to each other [9,10]. This may be due to heterogeneity in the primary tumor that comprises more than one cell clone [9]. Similarly, if a tumor progresses during oncologic treatment, we may see sufficient treatment effect on the dominant clone but synchronous growth of a minor clone that does not respond to the chosen therapy [8].

This heterogeneity of metastatic breast cancer has stimulated the development of new treatment approaches such as estrogen- and HER2-receptor targeting therapies. The prolonged survival of patients with metastatic breast cancer is strongly related to estrogen- and HER2-positive tumors and is most likely due to these targeting therapies [2,11]. Revolutionary approaches for future cancer treatment include therapies targeting specific receptors on cancer cells or genetic mutations in gene clusters that encode so-called ‘hub proteins’. As these mutations may be common across several cancer forms, future treatments could address clusters of genes or proteins rather than use the traditional categorization according to the originating tissue [12]. The next generation sequencing of tumor DNA and RNA appear to be relevant tools in future precision oncology as they might enable categorization of patients into different intervention ‘baskets’ with associated evidence-based recommendations for clinical management [13].

Precision medicine approaches, such as liquid biopsies that analyze circulating tumor DNA (ctDNA), may be able to accurately present the disease biology and potentially predict treatment response and prognosis [14]. However, potential drawbacks of sequencing ctDNA are a relatively low sensitivity and lack of visualization of the metastatic sites. These drawbacks could be reduced by the addition of molecular imaging to further support clinical decision-making. In the future, therefore, we expect that an optimal approach for response prediction and monitoring in metastatic cancers may be the combination of molecular imaging with genomic profiling using ctDNA.

## 3. Response Prediction and Response Monitoring

Most cancer treatments are still directed against the classic hallmarks of cancer and thus aim to suppress cancer growth and proliferation, cell division, and angiogenesis. While precision oncology approaches such as liquid biopsy [8,15] have created high expectations for the ability to predict treatment effect and monitor disease, the methodology for analyzing liquid biopsies and gene mutations is not yet available for clinical purposes. In the meantime, we need to address currently available methods to improve clinical decision-making for systemic treatment of metastatic breast cancer. A first step should be to use molecular imaging methods based on glycosylation in cancer [16].

Munkley et al. proposed that glycosylation is closely associated with all the hallmarks of cancer and that glycans play a major role in cancer growth [16]. A key feature is the shift from oxidative phosphorylation to aerobic glycolysis (the Warburg effect). The increased glucose uptake in cancer cells is used in the molecular imaging modality of FDG-PET/CT to enable prognosis and response monitoring of a range of oncologic diseases despite their varying molecular profiles. Another developing area involves theranostic approaches such as targeting radionuclides for combined PET-imaging and targeted radioisotope therapies. FDG-PET/CT imaging reflects glucose metabolism in cancer, and the level of FDG-uptake mirrors the aggressiveness of cancer cells [17]. Early papers have shown a clear relationship between FDG uptake (measured as the standardized uptake value, SUV) and disease prognosis—a relationship that was clearly demonstrated for malignant mesothelioma in 1999 [18]. The prognostic value of the maximum SUV (SUVmax) and the metabolic tumor volume (MTV) was also recently demonstrated for primary breast cancer [19], further supporting the central role of glycosylation in cancer. As recurrent and metastatic breast cancers are FDG-avid diseases [20,21], FDG-PET/CT may be a valuable tool for response monitoring. Morphologic response monitoring tools such as the Response Evaluation Criteria in Solid Tumors (RECIST) and RECIST 1.1 have traditionally been used for this purpose [3]. However, the emerging molecular-directed treatments mean that we need to consider a shift towards molecular-based criteria. A recent retrospective study of patients with metastatic breast cancer showed that response prediction of progression-free and disease-specific survival was superior with PERCIST compared to RECIST [22]. In the following, we present a case study to illustrate the potential of FDG-PET/CT and PERCIST in longitudinal response monitoring of metastatic breast cancer.

## 4. A Case Study of Longitudinal Response Monitoring in Metastatic Breast Cancer

A 47-year-old woman presented in January 2010 with locally advanced estrogen-positive, HER2-normal carcinoma in the right breast. She gave written informed consent for using her data for research purposes. Figure 1 shows FDG-PET/CT images at the time of diagnosis of primary breast cancer, where a metastasis in a small osteolytic lesion in her left trochanter major was also suggested. This was not visible with conventional X-rays, and thus neoadjuvant chemotherapy was initiated with cyclophosphamide/doxorubicin followed by docetaxel. The patient underwent mastectomy and axillary lymph node dissection and then received adjuvant radiotherapy and endocrine therapy. The small osteolytic lesion in the left trochanter major might have been overlooked or considered a benign lesion by a less sensitive reader. Today, when equivocal bone lesions are observed on FDG-PET/CT, it would be relevant to consider providing a better CT window for evaluation of bone structures or performing additional imaging such as Magnetic Resonance Imaging (MRI) or PET/MRI in order to increase the diagnostic accuracy [23].

Fifteen months later, the patient experienced lower back pain. As shown in Figure 2, FDG-PET/CT revealed a focal intense FDG uptake in the left trochanter major (A) whereas no changes were identified on CT. Bone metastasis was verified by biopsy. Treatment with capecitabine and ibandronate was initiated, and the treatment effect was monitored for the first year using contrast-enhanced CT. The last of four conventional CT scans detected an osteolytic lesion in vertebra L3, while FDG-PET/CT shortly after revealed unexpected massive progression to the skeleton (B). Treatment was changed to navelbine, and monitoring with FDG-PET/CT was started, using qualitative assessment of response. When progression occurred after four months (C), treatment was changed to eribulin, and subsequent disease regression was seen. The disease progressed again after five months (D) and treatment was altered to gemcitabine, resulting in several months without FDG-avid metastases. This sequence of progression, treatment change, and subsequent regression repeated itself, during which the patient received paclitaxel (2–3 months) (E); fulvestrant (5 months) (F); doxorubicin (10 months) (G); no treatment (4 months) (H), irinotecan (5 months) (I); and cyclophosphamide-methotrexate-fluorouracil (J). She passed away in February 2018.

In this patient case, conventional CT in the first year after initiation of systemic treatment for metastatic breast cancer showed complete response (CR) in three initial scans and then progressive disease (PD) on a fourth CT scan in April 2012. A supplemental FDG-PET/CT performed four days later revealed massive progression (Figure 2B; April 12), indicating that conventional CT detects progression later than FDG-PET/CT does. We need prospective trials, however, to investigate whether this could influence clinical decisions or patient survival and quality of life.

As the maximum intensity projection (MIP) images from scans are easily interpreted, they are a useful collaborative tool in clinical decision-making. Current guidelines emphasize that patients and their families should participate in decision-making processes, and MIP or ‘MIP-like’ images can be relevant here as well [4]. A further option is standardized semi-quantitative assessment using PERCIST. This may offer extra advantages and is addressed in the next section.

## 5. PERCIST 1.0 for Response Monitoring in Metastatic Breast Cancer

Figure 3 illustrates the treatment course of the patient in the case study based on PERCIST 1.0. We used the one-lesion method according to Wahl et al. [6] and O et al. [5] (Table 1) and thus used SULpeak of the hottest lesion, which may differ in consecutive scans. SULpeak represents the highest mean standardized uptake value (SUV) in a volume of interest of 1 cm^3^ corrected for lean body mass. The reference value in healthy liver tissue was measured in a volume of interest with 3 cm diameter to check for assessability in follow-up scans.

To illustrate some aspects related to the assessibility criteria listed in Table 1, we present data for the PERCIST response categorization in Table 2. In June 2013, we changed to a new scanner with matrix size 256 × 256 instead of 128 × 128. The ordered subset expectation maximization (OSEM) reconstruction algorithm was used for all scans. Although serum glucose levels were not available retrospectively, patients in our institution are asked to fast for six hours prior to the scan, so we consider this criterion to be met sufficiently. The injection-to-scan time was higher than 70 min on three occasions and differed on several occasions more than 15 min from baseline or pretreatment scans. The standard FDG dose applied at our institution is 4 MBq per kilogram, which we believe better accommodates potential weight changes than the dose proposed by PERCIST. SULmean in the liver differed more than 0.3 SUL units on several occasions (Table 2).

The software tool PET VCAR (AW server, version 3.2, Ext. 1.0, GE Healthcare, Waukesha, Milwaukee, WI, USA) was used for semi-automated measurement of the SULpeak for the hottest lesion at consecutive FDG-PET/CT scans. At the baseline scan in May 2011, SULpeak was 4.8 g/mL, SULmean in the liver was 1.5 g/mL, and the standard deviation of SULmean in the liver was 0.15 g/mL. This means that a SULpeak of a potential target lesion of 2.55 g/mL would have been sufficient for meeting measurability criteria of PERCIST 1.0. We used SULpeak values for the hottest lesion at consecutive scans, and PERCIST response categories were applied in all scans (Table 2).

We used the baseline or pretreatment scan as reference and, although it was not specified in PERCIST 1.0, we also used the nadir scan as reference in treatment intervals with decreasing SULpeak values in order to obtain clinically relevant response categories. As shown in Figure 3, we used color coding of response categories to graphically illustrate the long-term course. This approach can show decreasing or increasing trends in SULpeak before a change is detected by response categories or qualitative assessment. In doing this, we did not meet several specifications by PERCIST 1.0, but the depicted SULpeak fluctuations and PERCIST response categories appeared to mirror the disease activity in a timely manner. Thus, when PMD did not give rise to a change of treatment, we could still observe PMD on the following scan. When PMD did trigger a change of treatment, then PMR was observed four times (B–D, G), SMD once (E), and PMD three times (F, I, J), suggesting that the disease was more responsive to new treatment during its initial phase.

The graphical display provides an overview of treatment lines and impact of treatment change using SULpeak as a surrogate marker. A potential pitfall of the proposed PERCIST 1.0 is the lack of attention to the nadir level of SULpeak, which will be clinically relevant for monitoring cancer lesions that present with decreasing SULpeak values. However, this can be easily detected when using the graphical illustration (Figure 3).

Monitoring progression of metastatic cancer in this way has the potential to detect upcoming tumor clones that are resistant to ongoing oncologic treatment. Along with an increase or decrease of SULpeak in the target lesion, it can detect new suspicious lesions or unequivocal progression in non-target lesions, both of which would indicate progression. We consider PERCIST well suited for monitoring the biological processes in cancer progression and find it highly relevant for monitoring metastatic cancers such as breast cancer.

In the present case, it would have been valuable to retrospectively compare RECIST with PERCIST evaluation to estimate the degree of agreement/disagreement in response categorization [3,6]. Our FDG-PET/CT scans used low-dose CT, however, which prevented us from applying the RECIST evaluation. Our case study shows that the patient had ‘bone metastases only’ for a long time until June 2016, meaning that RECIST would predominantly have evaluated the disease as being non-measurable. It is our impression that the PERCIST criteria would have been suitable for response monitoring in this patient and potentially also in other patients with metastatic breast cancer. PERCIST appears to provide more information about the disease and its fluctuation, a higher degree of interrater agreement [24], a means to improve between-institution comparisons, and an integrative tool for shared clinical decision-making. Prospective studies are needed, however, to provide further clinical evidence.

## 6. Today

Current guidelines for management of advanced breast cancer recommend hormone-directed treatments as first-line therapy for estrogen- and HER2-receptor positive disease, whereas chemotherapy is recommended for receptor-negative disease. A biopsy of the metastatic lesion should thus be carried out before the patient starts treatment for metastatic breast cancer [4]. Evaluation of response to therapy is recommended every 2–4 months for endocrine therapy or after 2–4 cycles of chemotherapy. However, current guidelines do not recommend which imaging modality should be used for response evaluation [4].

A relevant issue to consider when using FDG-PET/CT for response evaluation or response monitoring in metastatic breast cancer is the optimal timing of PET/CT. Vach et al. concluded that schedule-optimizing studies should be performed [25]. While current guidelines do recommend regular time intervals for response evaluation, these should be reconsidered in light of the rapid shift in oncologic therapies and further development of imaging modalities [4,25].

A recent systematic review of PET/CT-based response evaluation [26] found that of 122 published papers, 112 (90%) were accuracy and/or prognostic studies. No randomized controlled trials were found, and only some studies included more than one post-baseline scan (31%). Further studies of response evaluation with PET/CT are thus needed to address issues of timing, repeat scans, and patient-relevant outcome measures. Well-planned multicenter trials will be required as large patient samples are needed to demonstrate significant outcomes.

Recent retrospective evidence showed that the PERCIST 1.0 one-lesion method was a better predictor of progression-free survival than RECIST [22]. The issue of the number of lesions to be quantified is relevant. In a recent study comparing the use of one-lesion versus five-lesion PERCIST for response prediction in metastatic breast cancer, Pinker et al. found no major impact on the prognostic value of the PERCIST approach [27]. A comprehensive review of the clinical impact of FDG-PET/CT for patients with breast cancer concluded that PET/CT is better at evaluating treatment response than conventional CT scan [28]. Several issues are still unanswered, however, and need further attention.

## 7. Tomorrow

There is no simple answer to the issue of optimal time interval in response monitoring for patients with metastatic breast cancer. The relevant interval will vary from cancer to cancer, from one therapy to another, from patient to patient. Imaging is indispensable, however, for characterizing the individual patient’s cancer and for choosing therapy.

Another question is how the individual scans should be compared to determine whether or not an actual change has occurred over time. PERCIST recommends comparing all successive scans to the baseline scan. This may not always be optimal, as reflected in our case study. In cases of metabolic regression, SULpeak will decrease and may have reached a nadir level that makes comparison to baseline misleading. In some cases, metabolic regression can be detected by comparison to the baseline scan, but in principle progression is observed by comparison to the nadir. This issue is relevant in clinical practice when several treatment lines are available, and especially with the development of targeting therapies. The nadir concept has previously been emphasized in the RECIST criteria [3].

A further issue is the standardization of acquisition and quantification procedures. This has been addressed by several authors in papers about PERCIST [5,6,29,30]. Scanners and reconstructions methods are under continuous technical improvement [29] to improve not only the quality of imaging but also the quantitative measures. We support the use of SULpeak as suggested in PERCIST 1.0, but we recognize that other measures and concepts are relevant and should be explored prospectively in future studies. These include total lesion glycolysis, metabolic tumor volume, the five-lesion method of PERCIST, and the use of PERCIST in initially non-measurable tumors. Other relevant explorative analyses would be to investigate the optimal limits of percentage increase and decrease for determining partial metabolic response or progressive metabolic disease. Limits of 30% have been suggested in PERCIST 1.0, but this needs to be confirmed in clinical trials.

There may be situations where qualitative evaluation of FDG-PET/CT results in a ‘mixed response’. This may refer to growth of a minor resistant clone alongside simultaneous response to a major and previously dominant clone. In such situations, PERCIST will most likely categorize this as progressive disease, consistent with the biology of cancer progression. The clinical decision for this may be equivocal, however, as it is very difficult to question the relevance of ongoing treatment that is having a positive effect on the initial dominant clone. If a shift in treatment is decided, however, this could potentially treat both clones. Clinical trials using patient-relevant outcome measures are thus highly relevant and should ideally include liquid biopsies.

## 8. Beyond

Artificial intelligence is an emerging field in which medical imaging such as FDG-PET/CT has an enormous potential by providing reproducible, observer-independent measures of cancer disease and its extent in the body. The development of radiomic tools for clinical decision support systems requires standardized and reproducible procedures, standardized data collection, and evaluation criteria [31]. Modelling of radiomic workflows is likely to be suitable for PERCIST and other analyses, providing a powerful tool in future medical treatment.

Another evolving field is the involvement of patients in treatment decisions, as emphasized in current clinical guidelines [4]. Shared decision-making facilitates patient engagement via collaboration between the clinician and the patient and includes elicitation of patient preferences and decision-making tools. Most patients with metastatic breast cancer want to be involved in decisions about their own treatment, and this appears to have positive effects [32,33]. Tools for shared decision-making in management of metastatic breast cancer have been developed, but recent papers call for more innovative and developmental research in the field [33,34]. None of the proposed tools include PET/CT scans or their results. As these play a crucial role in clinical decision-making, it is likely that greater communication between patients and health professionals about the timing and implications of scan results would contribute to shared decision-making.

In this perspective paper, we have dealt with FDG as the only PET tracer for response monitoring in metastatic breast cancer. Its huge importance is indisputable, not least because it typically shows uptake in different clones of breast cancer, and it is valuable even in long patient courses as the one described in our case study. However, it is worth investigating alternatives, especially more specific tracers that could characterize the individual patient and her cancer more accurately. Candidates include tracers targeting 18F-fluorotymidine, HER2, estrogen receptor, and gastrin-releasing peptide receptor [35,36,37,38]. Such specific tracers used alone or in combination may better predict treatment response. In a study by Ulaner et al., they found positive lesions on 89Zr-Trastuzumab PET/CT in six of 20 patients with HER2-negative primary breast cancer, and three of them had biopsy verified HER2-positive metastases [39]. This finding substantiates the theory of the heterogeneity of cancer and appoints opportunities for prediction of targeting therapy. Hence, when therapeutic ligands are developed, the theranostic potential represents a unique aspect of precision oncology that can benefit future patients [40].

Finally, the rapidly growing field of genomic analyses in precision oncology will without doubt become a major player in future research and clinical practice. We believe that molecular imaging will integrate with genomic analyses to provide improved treatment regimens that collaborative clinical teams can use for the benefit of patients with metastatic breast cancer.

## 9. Conclusions

Precision oncology is rapidly evolving, and targeted therapeutic options are already available for treatment of metastatic breast cancer. There is a gap between advanced modern therapy and contemporary monitoring of its effects, however, and current guidelines contain few recommendations for response evaluation criteria when monitoring response to oncologic treatment. The molecular imaging modality FDG-PET/CT may fill the gap as it reflects the central cancer hallmark of glycosylation, and it has proven clinical relevance in the diagnosis of metastatic breast cancer. There is evidence that the PET-based response evaluation criteria, PERCIST, can improve prediction in molecular-based treatment regimens compared to RECIST. We believe that PERCIST has the potential to provide greater information about disease fluctuation, thus better supporting clinical decision-making and patient empowerment.

## Figures and Tables

**Figure 1 cancers-11-01190-f001:**
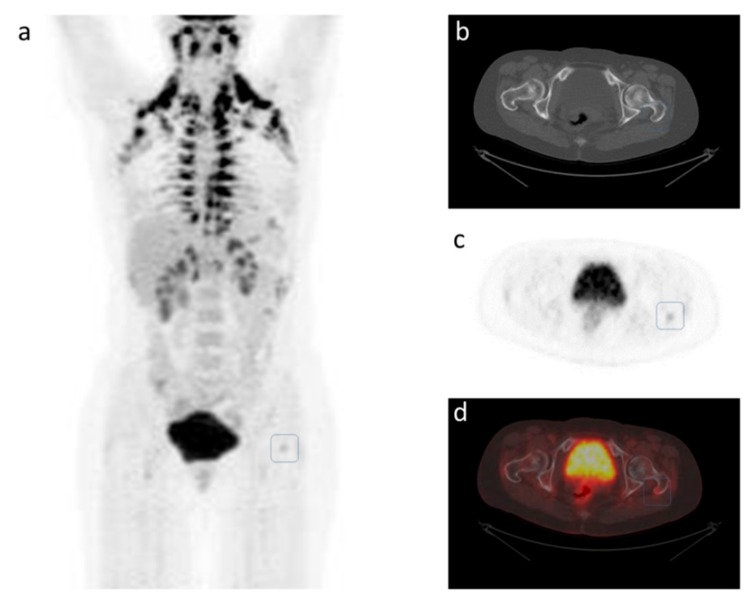
FDG-PET/CT performed for the first time in February 2010. Left (**a**): Maximum-intensity projection image showing an FDG-positive lesion in the left trochanter major, outlined by the blue square, suspicious of bone metastasis. High FDG uptake is seen in activated physiological brown fat tissue, but FDG uptake could not be seen in the primary tumor in the right breast. Right: Axial images of the pelvic region: (**b**) CT alone, (**c**) FDG-PET alone), and (**d**) fused FDG-PET/CT.

**Figure 2 cancers-11-01190-f002:**
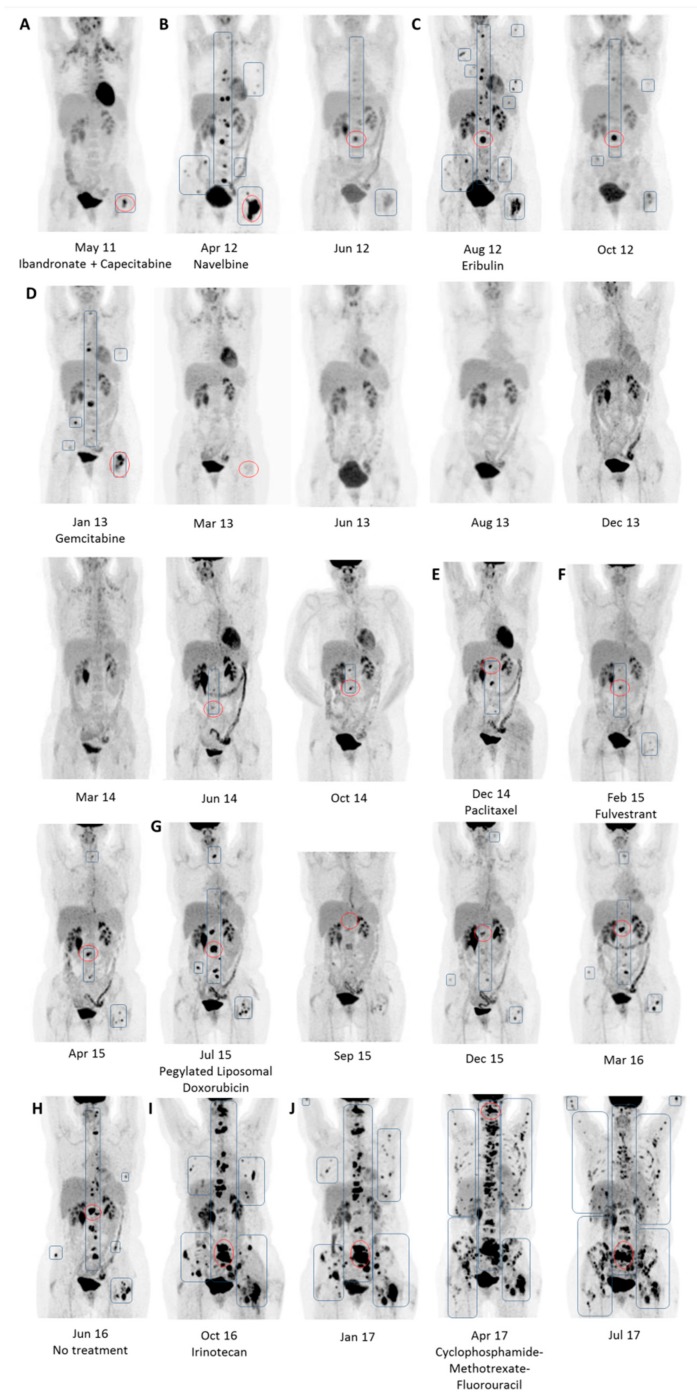
Maximum intensity projection images of a patient with metastatic breast cancer monitored longitudinally with FDG-PET/CT. Baseline scan (**A**) and pretreatment scans (**B**–**J**). Blue squares outline metastatic lesions. Red circles outline the hottest lesion representing a shifting target lesion for which SULpeak was measured using PERCIST 1.0.

**Figure 3 cancers-11-01190-f003:**
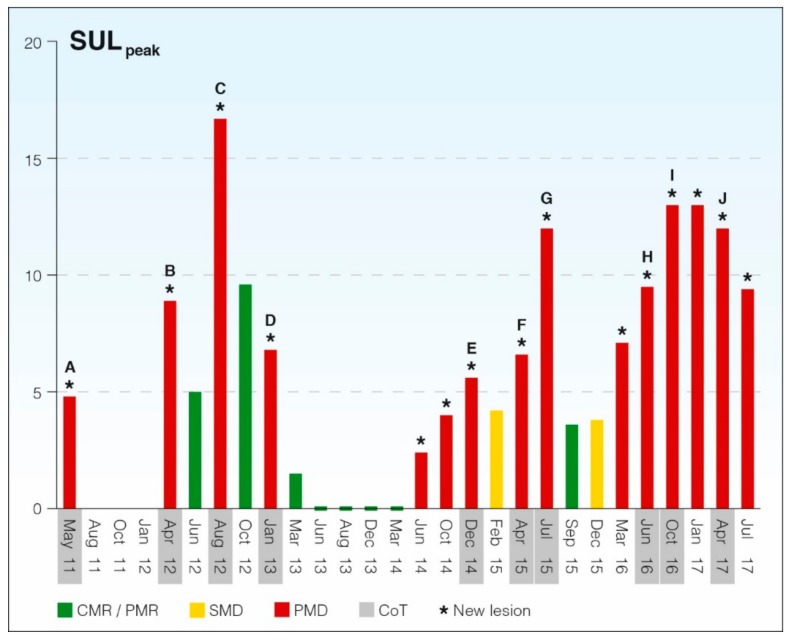
Graphical illustration of changes in PERCIST one-lesion SULpeak in a patient with metastatic breast cancer monitored for more than six years with FDG-PET/CT. Colored response categories: complete metabolic response (CMR), partial metabolic response (PMR), stable metabolic disease (SMD), and progressive metabolic disease (PMD). Grey-toned months represent time points for change of treatment (CoT). Conventional CT was performed Aug 11, Oct 11, and Jan 12. * Indicates scans with detection of new lesions.

**Table 1 cancers-11-01190-t001:** Specifications for the PERCIST 1.0 approach [5,6].

**Quantification Measure**	Suggested quantification measure	SUL_peak_ for the hottest tumor, which is not necessarily the same tumor on scans over time (one-lesion method)
	Suggested quantification measure	SUL_peak_ of up to five target lesions in hottest tumors, up to two per organ (five-lesions method)
**Measurability Criteria at Baseline**		Disease under study must be FDG-avid, meaning that SUL_peak_ should be ≥ 1 and ≥ 1.5 × SUL_mean, liver_ + 2 SD SUL_mean, liver_
**Assessabilty Criteria at Follow-Up**	Scanner criteria	Same scanner or scanner model at same site should be used
		Same reconstruction should be applied
		Properly calibrated scanners should be used
	Patient criteria	Patients should fast for at least 4 h
		Serum glucose levels should be < 200 mg/dL
	FDG criteria	Difference in injected dose should be ≤ 20%
		Difference in injection-to-scan-time between baseline and follow-up scan should be ≤ 15 min
		Injection-to-scan-time should be within 50–70 min
	Background activity criteria	Difference in SUL_mean, liver_ < 20% and < 0.3 SUL units
**Response Categories at Follow-Up**		Follow-up scans are suggested to be compared to the baseline/pretreatment scan
**Complete Metabolic Response**		Complete resolution of cancer-suspect lesions. Any previous lesion should have FDG uptake less than SUL_mean, liver_ and be indistinguishable from background. No new lesions appeared.
**Partial Metabolic Response**		Decrease in SUL_peak_ of hottest tumor ≥ 30% and at least 0.8 SUL units. No new lesions appeared. No increase in size greater than 30%. No unequivocal progression of a non-target lesion (increase in SUL_peak_ or size greater than 30%).
**Stable Metabolic Disease**		Increase or decrease of SUL_peak_ of less than 30%. No new lesions appeared. No increase in size greater than 30%. No unequivocal progression of a non-target lesion.
**Progressive Metabolic Disease**		Increase in SUL_peak_ of hottest tumor of ≥ 30% and of at least 0.8 SUL units, or development of one or more new lesions in a pattern suspect of cancer.

SUL_mean, liver_: mean standardized uptake value in liver, corrected for lean body mass; SUL_peak_: mean standardized uptake value in a one cm^3^ volume of interest; SD: standard deviation.

**Table 2 cancers-11-01190-t002:** Data for PERCIST response categorization for the treatment course presented.

Scan Month	Baseline/Pre- Treatment Scan (*)	SUL_mean, liver_	ITST/Min	Matrix Size	New Lesion	SUL_peak_ Target Lesion	Response Category
May 11	Yes (A)	1.51	64	128	Yes	4.78	BL
Apr 12	Yes (B)	2.03	60	128	Yes	8.89	PMD
Jun 12		1.66	82	128	No	4.99	PMR
Aug 12	Yes (C)	1.92	74	128	Yes	16.67	PMD
Oct 12		1.35	73	128	No	9.58	PMR
Jan 13	Yes (D)	1.68	61	128	Yes	6.81	PMD
Mar 13		1.68	64	128	No	1.47	PMR
Jun 13		1.85	68	128	No	-	CMR
Aug 13		1.92	58	256	No	-	CMR
Dec 13		2.07	58	256	No	-	CMR
Mar 14		1.80	111	256	No	-	CMR
Jun 14		1.95	58	256	Yes	2.42	PMD
Oct 14		1.93	67	256	Yes	4.03	PMD
Dec 14	Yes (E)	1.82	59	256	Yes	5.59	PMD
Feb 15		1.94	62	256	No	4.16	PMR
Apr 15	Yes (F)	2.04	66	256	Yes	6.59	PMD
Jul 15	Yes (G)	2.06	58	256	Yes	12.01	PMD
Sep 15		1.91	70	256	No	3.59	PMR
Dec 15		1.99	54	256	No	3.82	PMR
Mar 16		1.80	57	256	Yes	7.08	PMD
Jun 16	Yes (H)	1.81	66	256	Yes	9.46	PMD
Oct 16	Yes (I)	1.71	60	256	Yes	13.07	PMD
Jan 17		1.93	65	256	Yes	13.09	PMD
Apr 17	Yes (J)	1.44	64	256	Yes	11.93	PMD
Jul 17		2.03	61	256	Yes	9.44	PMD

BL: Baseline; PMD: progressive metabolic disease; PMR: partial metabolic response; CMR: complete metabolic response; * Treatment transitions A–J, as described in text; SUL_mean, liver_: mean standardized uptake value in liver, corrected for lean body mass (g/mL); ITST: injection to scan time; SUL_peak_: mean standardized uptake value in a 1 cm^3^ volume of interest (g/mL).

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
