# Peer review of "FDG-PET/CT for Response Monitoring in Metastatic Breast Cancer: Today, Tomorrow, and Beyond"

_cancers, 2019, doi:10.3390/cancers11081190_

Round 1

Reviewer 1 Report

The authors review the value of utilizing FDG PET/CT and PERCIST for response assessment in metastatic breast cancer, with the salient points demonstrated by an actual case. The images and detailed clinical info do an excellent job of delivering the message to non-imaging experts. I only have a few minor suggestions.

1. For the baseline PET/CT study, did the readers agree at the time of the imaging that the left femoral FDG uptake was due to metastasis, or was it after the fact? Perhaps you could explain that the left femoral uptake at baseline could have been considered a benign condition such as microfracture by a less sensitive reader? Or consider providing a better PET window image to show that it is a discrete FDG uptake lesion. 

2. Page 6 line 178/179. I believe the liver measurement at follow-up is not "to enable assessment", but rather "to check for assessability".

3. The assessment criteria in the table listing various scanner, patient,  etc criteria are not absolutely mandatory for PERCIST assessment. Some readers think one cannot perform PERCIST if these criteria are not met. I believe the purpose of the set is to ensure quality control. Not meeting a criterion does not mean the case should be automatically excluded from performing PERCIST assessment. 

4. Page 9 line 245. Consider adding reference demonstrating the high reader agreement. 

5. Page 10 line 285/286. The authors are correct in that PERCIST did a bad job clarifying which scan the comparison should be made with. The "baseline" study mentioned in PERCIST is not necessarily the very first FDG PET/CT scan, but actually directs to the "nadir" study just as the authors suggest. But yes, that fact is not delivered in the original manuscript of PERCIST 1.0.

6. Page 3 lines 127 to 131 and Table 1 paragraph alignments are off. 

7. Page 6 line 173 last name of the author for reference 5 is "O" and Joo Hyun is the first name. 

Author Response

Reviewer 1

English language and style are fine/minor spell check required.

Reply: We have addressed this, please see in cover letter.

Comments and Suggestions for Authors

The authors review the value of utilizing FDG PET/CT and PERCIST for response assessment in metastatic breast cancer, with the salient points demonstrated by an actual case. The images and detailed clinical info do an excellent job of delivering the message to non-imaging experts. I only have a few minor suggestions.

For the baseline PET/CT study, did the readers agree at the time of the imaging that the left femoral FDG uptake was due to metastasis, or was it after the fact? Perhaps you could explain that the left femoral uptake at baseline could have been considered a benign condition such as microfracture by a less sensitive reader? Or consider providing a better PET window image to show that it is a discrete FDG uptake lesion.

Reply: Thank you for the relevant considerations. The lesion was initially considered and reported as an osteolytic lesion suspect for metastasis by two experienced nuclear medicine physicians in our institution. We have changed the formulation (line 127) and added a sentence as suggested (lines 138-142).

Page 6 line 178/179. I believe the liver measurement at follow-up is not "to enable assessment", but rather "to check for assessability".

Reply: Thank you very much for pointing this out. We have rephrased the sentence accordingly into:  "to check for assessability"(line 180), and the term ‘assessability’ has also been inserted instead of ‘assessment’ in Table 1 and in line 198.

The assessment criteria in the table listing various scanner, patient, etc criteria are not absolutely mandatory for PERCIST assessment. Some readers think one cannot perform PERCIST if these criteria are not met. I believe the purpose of the set is to ensure quality control. Not meeting a criterion does not mean the case should be automatically excluded from performing PERCIST assessment.

Reply: Thank you for providing us this opinion which we share.  Accordingly, we have changed the terminology ‘requirement’ into ‘specification’ which has a less mandatory interpretation. This has been changed in line 220 and in the title for Table 1, page 7.

Page 9 line 245. Consider adding reference demonstrating the high reader agreement.

Reply: Yes, thanks, we have added a reference here. (Reference #24: Fledelius et al. Inter-observer agreement improves with PERCIST 1.0 as opposed to qualitative evaluation in non-small cell lung cancer patients evaluated with F-18-FDG PET/CT early in the course of chemo-radiotherapy. EJNMMI Res 2016, 6, 71)

Page 10 line 285/286. The authors are correct in that PERCIST did a bad job clarifying which scan the comparison should be made with. The "baseline" study mentioned in PERCIST is not necessarily the very first FDG PET/CT scan, but actually directs to the "nadir" study just as the authors suggest. But yes, that fact is not delivered in the original manuscript of PERCIST 1.0.

Reply: Thank you for the agreeing comment, we did therefore not make any changes to this.

Page 3 lines 127 to 131 and Table 1 paragraph alignments are off.

Reply: Alignment has been applied, thank you.

Page 6 line 173 last name of the author for reference 5 is "O" and Joo Hyun is the first name.

Reply: Thank you, this was changed here and in the reference list.

Reviewer 2 Report

This review present the potential of PERCIST criteria for monitoring response to therapy in breast cancer and highlight the interest of PERCIST over classic RESIST or RESIST1.1 classification.

In the figure one, the lesion in the left trochanter seems to have a low or moderated avidity for FDG. You should discuss the fact that in some cases the bony lesions can be better detected by SPECT/CT or the combination of SPECT/CT that PET-FDG and quote the Ref: Accuracy of whole-body HDP SPECT/CT, FDG PET/CT, and their combination for detecting bone metastases in breast cancer: an intra-personal comparison. O Rager, SA Lee-Felker, C Tabouret-Viaud, ER Felker, A Poncet, ... American journal of nuclear medicine and molecular imaging 8 (3), 159.

line 336. Discussing about other tracers, you could develop the discussion about the heterogeneotiy if the tumors, as discussed in the ref 33 you mention or in the Ref : 89Zr-Trastuzumab PET/CT for Detection of Human Epidermal Growth Factor Receptor 2–Positive Metastases in Patients With Human Epidermal Growth Factor Receptor 2–Negative Primary Breast Cancer. Ulaner, G.A. et al Clinical Nuclear Medicine 42(12):1

Author Response

Reviewer 2:

Extensive editing of English language and style required

Reply: We took care of this – please see text in the cover letter and throughout the revised manuscript.

This review present the potential of PERCIST criteria for monitoring response to therapy in breast cancer and highlight the interest of PERCIST over classic RESIST or RESIST1.1 classification.

In the figure one, the lesion in the left trochanter seems to have a low or moderated avidity for FDG. You should discuss the fact that in some cases the bony lesions can be better detected by SPECT/CT or the combination of SPECT/CT that PET-FDG and quote the Ref: Accuracy of whole-body HDP SPECT/CT, FDG PET/CT, and their combination for detecting bone metastases in breast cancer: an intra-personal comparison. O Rager, SA Lee-Felker, C Tabouret-Viaud, ER Felker, A Poncet, ... American journal of nuclear medicine and molecular imaging 8 (3), 159.

Reply: Thank you for the suggestion. Since the observed bone lesion was a small osteolytic lesion, we think MRI or PET/MRI would increase the sensitivity to a higher degree than HDP SPECT/CT in this case, and we have added a sentence addressing this (page 4, lines 137-140) as well as a relevant reference (Reference #23: Catalano, et al. Comparison of CE-FDG-PET/CT with CE-FDG-PET/MR in the evaluation of osseous metastases in breast cancer patients. Br J Cancer 2015, 112, 1452-1460.

line 336. Discussing about other tracers, you could develop the discussion about the heterogeneotiy if the tumors, as discussed in the ref 33 you mention or in the Ref : 89Zr-Trastuzumab PET/CT for Detection of Human Epidermal Growth Factor Receptor 2–Positive Metastases in Patients With Human Epidermal Growth Factor Receptor 2–Negative Primary Breast Cancer. Ulaner, G.A. et al Clinical Nuclear Medicine 42(12):1

Reply: Thank you for this suggestion. We have added a few lines addressing this issue, substantiating the evidence on heterogeneity of tumors.
